# Diagnostic performance of congestion score index evaluated from chest radiography for acute heart failure in the emergency department: A retrospective analysis from the PARADISE cohort

Masatake Kobayashi[1☉], Amine Douair[2☉], Kevin Duarte[1], Déborah Jaeger[1,2], Gaetan Giacomin[2], Adrien Bassand[2], Victor Jeangeorges[2], Laure Abensur Vuillaume[3], Gregoire Preud'homme[1], Olivier Huttin[1], Faiez Zannad[1], Patrick Rossignol[1], Tahar Chouihed[1,2], Nicolas Girerd[1]*

1 Université de Lorraine, Inserm, Centre d'Investigations Cliniques-1433, and Inserm, CHRU Nancy, F-CRIN INI-CRCT, Nancy, France, 2 Emergency Department, University Hospital of Nancy, Vandoeuvre-les-Nancy, France, 3 Emergency Department, Hospital of Metz Thionville, Ars-Laquenexy, France

☉ These authors contributed equally to this work.
* n.girerd@chru-nancy.fr

**Data Availability Statement:** Under French law, deidentified data cannot be transferred to someone

## Abstract

### Background

Congestion score index (CSI), a semiquantitative evaluation of congestion on chest radiography (CXR), is associated with outcome in patients with heart failure (HF). However, its diagnostic value in patients admitted for acute dyspnea has yet to be evaluated.

### Methods and findings

The diagnostic value of CSI for acute HF (AHF; adjudicated from patients' discharge files) was studied in the Pathway of dyspneic patients in Emergency (PARADISE) cohort, including patients aged 18 years or older admitted for acute dyspnea in the emergency department (ED) of the Nancy University Hospital (France) between January 1, 2015 and December 31, 2015. CSI (ranging from 0 to 3) was evaluated using a semiquantitative method on CXR in consecutive patients admitted for acute dyspnea in the ED. Results were validated in independent cohorts (N = 224). Of 1,333 patients, mean (standard deviation [SD]) age was 72.0 (18.5) years, 686 (51.5%) were men, and mean (SD) CSI was 1.42 (0.79). Patients with higher CSI had more cardiovascular comorbidities, more severe congestion, higher b-type natriuretic peptide (BNP), poorer renal function, and more respiratory acidosis. AHF was diagnosed in 289 (21.7%) patients. CSI was significantly associated with AHF diagnosis (adjusted odds ratio [OR] for 0.1 unit CSI increase 1.19, 95% CI 1.16–1.22, $p$ < 0.001) after adjustment for clinical-based diagnostic score including age, comorbidity burden, dyspnea, and clinical congestion. The diagnostic accuracy of CSI for AHF was >0.80, whether alone (area under the receiver operating characteristic curve [AUROC] 0.84, 95%

not authorized by the CNIL (Commission Nationale de l'Informatique et des Libertés) to perform the analysis. To comply with this national data regulation, we will provide access to the data on a secured server held by our institution upon reasonable request to the primary investigator of the study. Importantly, Nancy CIC-P works with a number of international groups using this secured server. Requests can be made at the following address: cic@chru-nancy.fr.

**Funding:** MK is granted by the RHU Fight-HF, a public grant overseen by the French National Research Agency (ANR) as part of the second "Investissements d'Avenir" program (ANR-15-RHUS-0004). MK, PR,KD, NG, and FZ are supported by the RHU Fight-HF, a public grant overseen by the French National Research Agency (ANR) as part of the second "Investissements d'Avenir" program (ANR-15-RHUS-0004) and by the FrenchPIA project "Lorraine Université d'Excellence" (ANR-15-IDEX-04-LUE). The funders had no role in study design, data collection and analysis, decision to publish, or preparation of the manuscript.

**Competing interests:** We have read the journal's policy and the authors of this manuscript have the following competing interests: NG receives honoraria from Novartis and Boehringer. TC receives fees from Novartis for scientific board. FZ and PR are the cofounders of CardioRenal. FZ reports personal fees from Boehringer Ingelheim, Janssen, Novartis, Boston Scientific, Amgen, CVRx, AstraZeneca, Vifor Fresenius, Cardior, Cereno pharmaceutical, Applied Therapeutics, Merck, Bayer and Cellprothera, and is a founder of Cardiovascular Clinical Trialists. PR reports grants and personal fees from AstraZeneca, Bayer, CVRx, Fresenius, and Novartis, personal fees from Grunenthal, Servier, Stealth Peptides, Vifor Fresenius Medical Care Renal Pharma, Idorsia, NovoNordisk, Ablative Solutions, G3P, Corvidia and Relypsa. Other co-authors have declared that no competing interests exist.

**Abbreviations:** AHF, acute heart failure; AUC, area under the curve; AUROC, area under receiver operating characteristic curve; BMI, body mass index; BNP, b-type natriuretic peptide; CNIL, Commission Nationale de l'Informatique et des Libertés; COPD, chronic obstructive pulmonary disease; CSI, congestion score index; CXR, chest radiography; ED, emergency department; eGFR, estimated glomerular filtration rate; ESC, European Society of Cardiology; HF, heart failure; ICALOR, Insuffisance CArdiaque en LORraine; NRI, net reclassification improvement; OR, odds ratio; PARADISE, Pathway of dyspneic patients in

CI 0.82–0.86) or in addition to the clinical model (AUROC 0.87, 95% CI 0.85–0.90). CSI improved diagnostic accuracy on top of clinical variables (net reclassification improvement [NRI] = 94.9%) and clinical variables plus BNP (NRI = 55.0%). Similar diagnostic accuracy was observed in the validation cohorts (AUROC 0.75, 95% CI 0.68–0.82). The key limitation of our derivation cohort was its single-center and retrospective nature, which was counter-balanced by the validation in the independent cohorts.

## Conclusions

In this study, we observed that a systematic semiquantified assessment of radiographic pulmonary congestion showed high diagnostic value for AHF in dyspneic patients. Better use of CXR may provide an inexpensive, widely, and readily available method for AHF triage in the ED.

## Author summary

### Why was this study done?

- Chest radiography (CXR) is often performed in patients admitted for acute dyspnea in the emergency department (ED); however, its assessment is generally not standardized.

- Latest guidelines for heart failure (HF) emphasize the limitations of CXR in its current use.

- A standardized approach to quantify pulmonary congestion from CXR could potentially improve its diagnostic performance for acute HF (AHF).

### What did the researchers do and find?

- We studied the diagnostic value of a semiquantitative approach (the congestion score index [CSI]) to pulmonary congestion on CXR for AHF in a retrospective cohort study of 1,333 patients with acute dyspnea admitted to the ED.

- This CSI was significantly associated with AHF diagnosis.

- The CSI also improved diagnostic accuracy over clinical parameters with or without inclusion of natriuretic peptide.

- This good diagnostic accuracy of the CSI was externally validated in independent cohorts ($N = 224$).

### What do these findings mean?

- The quantification of radiographic pulmonary congestion using the CSI improved diagnostic accuracy for AHF on top of clinical parameters and natriuretic peptide.

Emergency; PURPLE, Pathway and Urgent caRe of Dyspneic Patient at the Emergency Department in LorrainE District; ROC, receiver operating characteristics; SD, standard deviation; TRIPOD, Transparent Reporting of a multivariable prediction model for Individual Prognosis Or Diagnosis.

## Introduction

Acute heart failure (AHF) is 1 of the leading causes of acute dyspnea in the emergency department (ED) [1] and is associated with a higher risk of morbidity and mortality [2,3]. In-hospital mortality is reported to be greater than 10% [4] and has remained stable in the last 30 years. As prognosis is associated with initiation time of specific therapies [5], current guidelines emphasize the importance of early diagnosis and treatment initiation to improve clinical outcomes [5,6]. However, a minority of patients with AHF receive treatment within 1 hour of admission [5], in contradiction with current recommendations [6]. In addition, a third of AHF diagnosis are missed in the ED [6], further delaying access to care. An increasing number of better diagnostic tools for AHF are available in the ED [7–9]. However, there is likely room for improving the diagnostic approach to AHF from widely available routine tools including chest radiography (CXR).

CXR is a fast and inexpensive method performed systematically in the ED in patients with acute dyspnea [10–12]. It is the first-line diagnostic imaging modality advocated in current guidelines [13]. However, its diagnostic accuracy for HF has been reported to be relatively low [14–16]. In particular, diagnosing HF in patients with concomitant lung diseases such as chronic obstructive pulmonary disease (COPD) and pneumonia still remains a challenge [17].

A new semiquantitative approach to pulmonary congestion has recently emerged in the field of HF. Congestion score index (CSI) is a semiquantitative approach to pulmonary congestion based on a 6-zone evaluation of CXR, scoring each zone from 0 (no congestion) to 3 (intense alveolar pulmonary edema). CSI is a strong risk stratifier in patients with stable or worsening HF [18–20]. However, there are little available data regarding its diagnostic value for AHF.

The aims of the present study are to investigate the diagnostic value of pulmonary congestion assessed with CSI for AHF in patients admitted for acute dyspnea in the ED and to assess its discriminative value comparatively to and on top of currently used clinical diagnostic models and natriuretic peptide measurements.

## Methods

### Study population

This study is reported as per the Transparent Reporting of a multivariable prediction model for Individual Prognosis Or Diagnosis (TRIPOD) guideline (S1 Checklist). The Pathway of dyspneic patients in Emergency (PARADISE) cohort is a retrospective cohort study including consecutive patients aged 18 years or older who were admitted for acute dyspnea in the ED of the Nancy University Hospital (France) between January 1, 2015 and December 31, 2015 as detailed previously [21,22]. The hospital's electronic charts (resurgences) were systematically reviewed by investigators to search for the records of all patients admitted for acute dyspnea in the ED. All patients with signs or symptoms of dyspnea or requiring oxygen therapy for dyspnea during (or prior to) their ED stay were included. Patients with shock or cardiac arrest were excluded from this analysis as dyspnea was not the primary condition triggering ED admission. The PARADISE cohort is consequently a cohort of unselected consecutive patients with acute dyspnea in the ED. In the current study, a total of 1,333 dyspneic patients with available information on CXR at the ED were analyzed (S1 Fig). Demographic parameters, medical history, physical examination, laboratory findings, and treatment received in the ED were retrieved from the patients' electronic records.

External validation was performed on the merged dataset from the HF disease management program entitled "Insuffisance CArdiaque en LORraine (ICALOR)" [23,24] (included at a

different time period that the data from the Nancy University Hospital used in the derivation set) and data from the Epinal Hospital (a secondary care hospital located 70 km from the Nancy University Hospital) within the Pathway and Urgent caRe of Dyspneic Patient at the Emergency Department in LorrainE District (PURPLE) multicenter cohort (NCT03194243). Briefly, we used the data from the previously described [19] 117 patients included in the ICA-LOR disease management cohort during a hospitalization for acutely decompensated HF in the Cardiology Department of the Nancy University Hospital (France). We also used the data of 107 consecutive patients admitted for acute dyspnea with available CXR data and discharge diagnosis in the ED of the Epinal Hospital. This external validation set allowed us to test CSI in a significantly different setting (mixing cardiology department from a tertiary care hospital and an ED of a secondary care hospital) than our derivation set.

Under French law, no formal Institutional Review Board approval is required for data extraction from patient medical records in single-center cohorts (the PARADISE and ICALOR cohorts). For the PURPLE cohort, patients were informed through a notice at admission and could refuse their inclusion in the study, although no formal consent was required in keeping with the framework of the Commission Nationale de l'Informatique et des Libertés (CNIL). The PARADISE cohort was recorded by the local hospital corresponding agent of the CNIL (Number R2016-08) and was registered on clinicaltrials.gov (NCT02800122). The PURPLE cohort was approved by an ethical board ("Comité de Protection des personnes"—Number 2016–63, ID RCB 2016-A01877-44) and CNIL (Number DR-2017-098) and was registered on clinicaltrials.gov (NCT03194243).

## Diagnosis of heart failure

HF was diagnosed according to the European Society of Cardiology (ESC) guidelines [25]. Diagnosis of AHF was coded independently by 2 medical physicians (GG and TH) according to the ESC guidelines [25]. Each physician had access to all ED medical charts as well as additional hospital admission test results and records (e.g., echocardiography, natriuretic peptide level, and patient response to diuretic/bronchodilator therapy) but were blinded to CSI quantification on CXR. Homogenous coding was ensured by a trained senior physician (TC). Importantly, to determine the discharge diagnosis, we focused on the main cause of acute dyspnea rather than background medical history or coexisting conditions.

## Radiographic congestion score index

Radiographic CSI was used to quantify the severity of pulmonary congestion in CXR as previously published [18,19]. After dividing the lung field into 6 topographical zones, each area was assessed as follows: Score 0, no congestion sign; Score 1, cephalization (superior area), perihilar haze or perivascular/peribronchial cuffing, or Kerley A lines (middle area), Kerley B, or C lines (inferior area); Score 2, interstitial or localized/mild alveolar pulmonary edema; Score 3, intense alveolar pulmonary edema (Fig 1). To enhance the reproducibility of the severity of confluent edema, a portion of the divided lung fields which was visually similar to the cardiac silhouette was regarded as an intense zone, whereas the field with weaker density was regarded as a mildly intense zone. Lung areas were not scored when more than one-third of the divided lung fields were occupied by pleural effusion (including vanishing tumor), atelectasis, or cardiac silhouette. CSI was calculated as the sum of the scores in each zone divided by the number of available zones. An examiner also assessed the presence of pneumonia, pleural effusion, cardiomegaly by cardiothoracic ratio (>50%), and the difficulty in assessing CSI.

CXR was analyzed by a single emergency physician (AD), blinded to clinical data and discharge diagnosis, with no previous training in congestion quantification on CXR prior to that

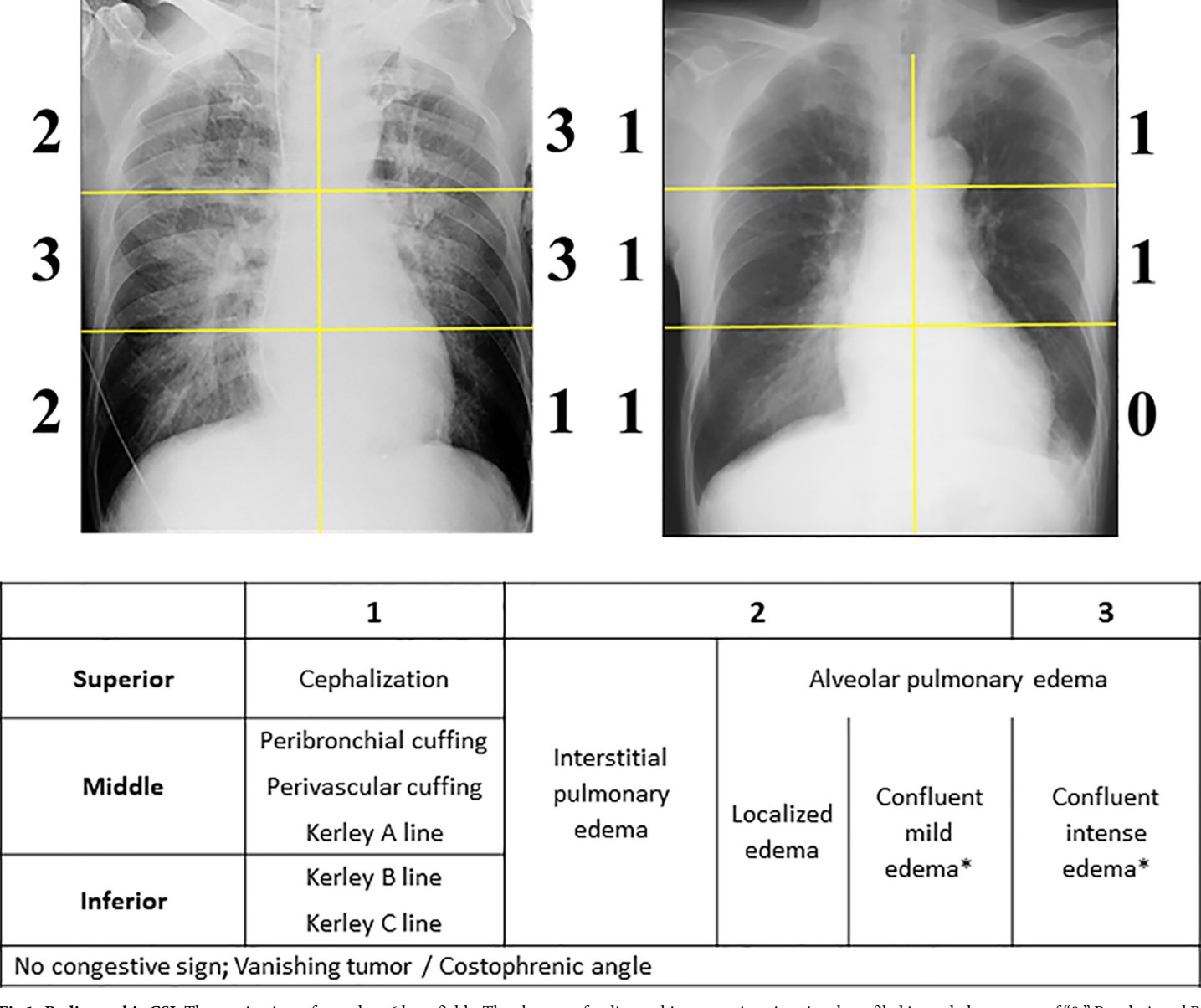

**Fig 1. Radiographic CSI.** The scoring is performed on 6 lung fields. The absence of radiographic congestion signs in a lung filed is graded as a score of "0." Panels A and B provide examples. (A) Example: CSI = (2+3+3+3+2+1)/6 = 2.33. There is diffuse alveolar edema, appearing as intense edema in the left superior field and middle fields. (B) Example: CSI = (1+1+1+1+1+0)/6 = 0.83. Cephalization in superior fields and peribronchial and perivascular cuffing are visible in middle fields, respectively. *Confluent edema was regarded as intense edema when the density in an area of the divided lung field was visually similar to that of cardiac silhouette. CSI, congestion score index.

in the present study. After a short training (approximately 3 hours) using a 20-patient sample with a radiographic CSI expert (MK), intraobserver and interobserver agreements (with MK) were tested on 30 randomly selected patients, while blinded to clinical status and diagnosis. Intraclass correlation coefficients showed good reproducibility (intraobserver agreement, 0.85, 95% CI 0.71 to 0.93 and interobserver agreement 0.81, 95% CI 0.64 to 0.90).

### Brest score

The Brest score was calculated for every patient based on the patients' medical charts. This diagnostic score for AHF in dyspneic patients is based on age, comorbidities (i.e., prior history of HF, myocardial infarction, and COPD), pattern of dyspnea, ST segment abnormalities, and signs/symptoms of congestion (i.e., rales and leg edema) [26,27].

### Statistical analysis

Categorical variables are expressed as frequencies (percentages) and continuous variables as means ± standard deviation (SD) or median (25th and 75th percentiles) according to the distribution of the variables. Comparisons of demographic, clinical, and biological parameters among quartiles of radiographic CSI were analyzed using $\chi^2$ tests for categorical variables and ANOVA or Kruskal–Wallis test for continuous variables. Interobserver and intraobserver agreements of CSI were assessed with the intraclass correlation coefficient.

We analyzed 1,333 dyspneic patients with 289 AHF diagnosis—which provided a sizeable statistical power to assess diagnostic performance in a multivariable setting [28]. A logistic regression model was used to assess the association of CSI with diagnosis of AHF. Multivariable analyses included relevant confounders as previously shown [21]: model 1: age, sex, body mass index (BMI), presence of hypertension, diabetes mellitus, coronary artery disease, atrial fibrillation, prior HF admission, use of angiotensin converting enzyme inhibitor/angiotensin receptor blocker, beta-blocker, diuretics, leg edema, jugular venous distension, hemoglobin, white blood cell count, and estimated glomerular filtration rate (eGFR, as calculated by the Chronic Kidney Disease Epidemiology Collaboration formula [29]) at admission; model 2: Brest score. Receiver operating characteristics (ROC) curve was used to determine the diagnostic value of CSI in AHF. All correlation coefficients of variables included in the models were less than 0.50 with CSI, suggesting the absence of important in-model collinearity.

The increase in discriminative value of the addition of CSI for AHF diagnosis on top of the aforementioned potential covariates was assessed using continuous net reclassification improvement (NRI) [30]. In addition, in 498 (37.4%) patients who had available b-type natriuretic peptide (BNP) measurements, the added value of CSI on the top of the Brest score and BNP was assessed.

This analysis on the PARADISE cohort was planned in February 2019, although no formal analysis was written. The general analysis intention was to evaluate the diagnostic value of CSI for AHF. Of note, among all patients with available data, CXR was assessed, blinded for clinical data and diagnosis, and standard statistical approaches were used. Based on recommendations made during the peer-review process, we conducted restricted cubic spline regression analysis for the association between CSI and AHF diagnosis.

All analyses were performed using R version 3.4.0 (R Development Core Team, Vienna, Austria). A 2-sided $p$-value $< 0.05$ was considered statistically significant. No imputation was performed.

## Results

### Baseline characteristics

Less than 10% of the population had no available data on CXR (S1 Table). These patients were markedly younger and had less comorbidities than patients who underwent CXR. The characteristics of the PARADISE cohort population across different discharge diagnoses such as AHF, COPD, and pneumonia are depicted in S2 Table.

In a total of 1,333 patients included in this study, a half were men, mean age was $72.0 \pm 18.5$ years, mean BMI was $25.5 \pm 5.5$ kg/m$^2$, and less than 10% had a prior admission for HF (7.1%) (Table 1). CXR was considered as difficult to interpret during CXR reviewing in 502 patients (37.7%). Mean CSI was $1.42 \pm 0.79$. Patients with a higher CSI had more cardiovascular risk factors, comorbidities, more frequent prior HF admission, more severe congestion, inflammation status, higher BNP, poorer renal function, and more respiratory acidosis at admission (Table 1).

## Association of congestion score index with adjudicated discharge diagnosis of acute heart failure

In this study, 289 (21.7%) patients were diagnosed with AHF at discharge. Higher CSI was significantly associated with AHF diagnosis (odds ratio [OR] [95% CI] for a 0.1 unit increase in CSI = 1.22 [1.19 to 1.25], $p < 0.001$) even after adjustment for potential clinical confounders (adjusted OR [95% CI] in CSI = 1.18 [1.15 to 1.22], $p < 0.001$) and the Brest score (adjusted OR [95% CI] in CSI = 1.19 [1.16 to 1.22], $p < 0.001$). The association of CSI with AHF diagnosis using restricted cubic spline regression analysis is shown in Fig 2. CSI had a linear association with AHF diagnosis ($p > 0.05$ for nonlinearity), and higher CSI showed an increased risk of AHF diagnosis (OR [95% CI] for CSI score 1.0 = 4.09 [2.50 to 6.71], $p < 0.001$; OR [95% CI] for CSI score 1.5 = 13.92 [6.32 to 30.65], $p < 0.001$; OR [95% CI] for CSI score 2.0 = 37.03 [16.09 to 85.26], $p < 0.001$—considering CSI score 0.5 as a reference). Similar results were observed after adjustment for the Brest score (adjusted OR [95% CI] for CSI score 1.0 = 3.26 [1.93 to 5.48], $p < 0.001$; adjusted OR [95% CI] for CSI score 1.5 = 9.07 [4.00 to 20.59], $p < 0.001$; adjusted OR [95% CI] for CSI score 2.0 = 20.88 [8.83 to 49.35], $p < 0.001$—considering CSI score 0.5 as a reference) (Fig 2).

## Diagnostic value of the congestion score index

In the whole population, CSI exhibited high discrimination for AHF as reflected by an area under the curve (AUC) of 0.84 (0.82 to 0.86) (Figs 3 and 4). Similarly, high AUC was observed across subgroups of age, sex, and comorbidities (obesity and COPD) or associated diagnosis (pneumonia and pleural effusion) (Fig 3). In contrast, subgroups without cardiomegaly assessed by CXR had higher AUC compared to those with cardiomegaly (AUC [95% CI] = 0.85 [0.81 to 0.89] versus 0.75 [0.70 to 0.79], respectively). In addition, the diagnostic value of CSI was influenced by the patient's position (AUC [95% CI] = 0.83 [0.80 to 0.89] in sitting position and 0.79 [0.73 to 0.84] in supine position) as well as the difficulty in assessing CSI (AUC [95% CI] = 0.92 [0.86 to 0.97] for easy, 0.84 [0.80 to 0.88] for moderate, and 0.80 [0.75 to 0.84] for difficult assessments).

AUC for the Brest score was 0.78 [0.75 to 0.81]. The combination of CSI and Brest score yielded a high AUC for AHF (AUC [95% CI] = 0.87 [0.85 to 0.90]).

## Improvement in reclassification associated with acute heart failure diagnosis

The addition of CSI on top of the Brest score significantly improved reclassification of AHF diagnosis (NRI [95% CI] = 94.9 [83.5 to 106.2], $p < 0.001$) (Fig 4).

Furthermore, in patients with available BNP data ($N = 496$), CSI still significantly improved reclassification of AHF diagnosis on top of BNP and the Brest score (NRI [95% CI] = 55.0 [38.0 to 72.0], $p < 0.001$, delta AUC [95% CI] = 2.9 [0.6 to 5.2], $p = 0.015$) (Figs 4 and 5). The diagnostic value of the joint use of the Brest score and CSI was not significantly different than that of the joint use of the Brest score and BNP (NRI [95% CI] = 4.4 [−13.3 to 22.1], $p = 0.63$, delta AUC [95% CI] = −1.4 [−5.0 to 2.1], $p = 0.42$). In this subgroup, the Brest score had a moderate

**Table 1. Baseline characteristics according to the radiographic CSI (quartiles).**

| | Global (N = 1,333) | Quartile I, <0.84 (N = 376) | Quartile II, 0.84–1.40 (N = 328) | Quartile III, 1.40–2.00 (N = 359) | Quartile IV, ≥2.00 (N = 270) | % missing value | p-value | Adjusted p-value* |
|---|---|---|---|---|---|---|---|---|
| | | | **CSI quartiles** | | | | | |
| Age, years | 72.0 ± 18.5 | 57.4 ± 20.9 | 73.4 ± 15.4 | 79.3 ± 12.6 | 80.9 ± 11.8 | 0 | **<0.001** | — |
| Men, N (%) | 686 (51.5%) | 212 (56.4%) | 168 (51.2%) | 166 (46.2%) | 140 (51.9%) | 0 | 0.06 | 0.26 |
| BMI, kg/m$^2$ | 25.5 ± 5.5 | 24.7 ± 4.8 | 25.3 ± 5.0 | 25.5 ± 5.8 | 26.8 ± 6.4 | 1.1 | **<0.001** | — |
| Medical history, N (%) | | | | | | | | |
| Hypertension | 729 (54.7%) | 111 (29.5%) | 194 (59.1%) | 236 (65.7%) | 188 (69.6%) | 0 | **<0.001** | **0.03** |
| Diabetes mellitus | 302 (22.7%) | 50 (13.3%) | 62 (18.9%) | 94 (26.2%) | 96 (35.6%) | 0 | **<0.001** | **0.03** |
| Dyslipidemia | 280 (21.0%) | 51 (13.6%) | 79 (24.1%) | 82 (22.8%) | 68 (25.2%) | 0 | **<0.001** | 0.36 |
| Coronary artery disease | 162 (12.2%) | 16 (4.3%) | 36 (11.0%) | 63 (17.5%) | 47 (17.4%) | 0 | **<0.001** | **0.001** |
| Atrial fibrillation | 313 (23.5%) | 33 (8.8%) | 70 (21.3%) | 107 (29.8%) | 103 (38.1%) | 0 | **<0.001** | **0.003** |
| HF | 258 (19.4%) | 21 (5.6%) | 47 (14.3%) | 93 (25.9%) | 97 (35.9%) | 0 | **<0.001** | **<0.001** |
| Prior HF admission, N (%) | 95 (7.1%) | 6 (1.6%) | 15 (4.6%) | 28 (7.8%) | 46 (17.0%) | 0 | **<0.001** | **<0.001** |
| Medication, N (%) | | | | | | | | |
| ACEi/ARB | 444 (34.7%) | 67 (18.3%) | 112 (36.5%) | 142 (40.8%) | 123 (47.7%) | 4.1 | **<0.001** | **0.02** |
| Beta-blocker | 306 (23.9%) | 45 (12.3%) | 70 (22.8%) | 103 (29.6%) | 88 (34.1%) | 4.1 | **<0.001** | **0.047** |
| Spironolactone | 64 (5.0%) | 16 (4.4%) | 13 (4.2%) | 19 (5.5%) | 16 (6.2%) | 4.1 | 0.65 | 0.88 |
| Diuretics | 368 (28.8%) | 40 (10.9%) | 81 (26.4%) | 127 (36.5%) | 120 (46.5%) | 4.1 | **<0.001** | **<0.001** |
| Calcium channel blocker | 258 (20.2%) | 24 (6.6%) | 67 (21.8%) | 82 (23.6%) | 85 (32.9%) | 4.1 | **<0.001** | **<0.001** |
| Statin | 300 (23.5%) | 58 (15.8%) | 81 (26.4%) | 87 (25.0%) | 74 (28.7%) | 4.1 | **<0.001** | 0.58 |
| O2 flow, L/min | 4.0 (2.0–9.0) | 3.0 (2.0–9.0) | 3.0 (2.0–9.0) | 3.0 (2.0–9.0) | 5.0 (3.0–9.0) | 53.0 | **0.005** | **0.001** |
| Physical examination, N (%) | | | | | | | | |
| Leg edema | 334 (25.1%) | 28 (7.4%) | 65 (19.8%) | 117 (32.6%) | 124 (45.9%) | 0 | **<0.001** | **<0.001** |
| Jugular venous distension | 43 (3.3%) | 4 (1.1%) | 8 (2.4%) | 12 (3.4%) | 19 (7.3%) | 1.4 | **<0.001** | **0.04** |
| Rales | 454 (35.2%) | 63 (17.3%) | 104 (32.3%) | 148 (42.8%) | 139 (54.5%) | 3.4 | **<0.001** | **<0.001** |
| Systolic BP, mmHg | 132.1 ± 26.0 | 129.6 ± 22.7 | 132.3 ± 26.0 | 132.9 ± 26.8 | 134.5 ± 28.9 | 0.1 | 0.13 | 0.98 |
| Diastolic BP, mmHg | 73.5 ± 17.6 | 77.2 ± 16.3 | 72.4 ± 17.8 | 71.6 ± 18.1 | 72.4 ± 18.0 | 0.1 | **<0.001** | 0.40 |
| Heart rate, bpm | 95.7 ± 20.7 | 98.1 ± 19.0 | 94.7 ± 20.2 | 94.2 ± 20.1 | 95.5 ± 23.8 | 0.4 | **0.02** | 0.32 |
| Respiratory rate, /min | 26.3 ± 7.9 | 24.5 ± 7.6 | 26.5 ± 7.5 | 26.8 ± 8.0 | 27.5 ± 8.4 | 31.3 | **<0.001** | 0.19 |
| CSI | 1.4 ± 0.8 | 0.5 ± 0.3 | 1.2 ± 0.2 | 1.8 ± 0.2 | 2.5 ± 0.3 | 0 | **<0.001** | **<0.001** |
| Laboratory findings | | | | | | | | |
| Hemoglobin, g/dl | 12.8 ± 2.0 | 13.5 ± 1.9 | 12.8 ± 2.0 | 12.6 ± 2.0 | 12.2 ± 2.0 | 0.3 | **<0.001** | **<0.001** |
| White blood cell count, μ/l | 11,300 (8,300–15,400) | 10,765 (8,420–14,100) | 10,900 (7,700–15,100) | 11,575 (7,800–15,600) | 12,600 (8,900–16,700) | 2.0 | **0.006** | **0.007** |
| C-reactive protein, mg/dl | 6.6 (1.8–14.0) | 3.9 (0.8–10.5) | 7.7 (2.3–14.8) | 7.7 (2.3–14.5) | 7.1 (2.1–15.4) | 7.7 | **<0.001** | **<0.001** |
| Sodium, mmol/l | 136.9 ± 5.3 | 137.1 ± 4.9 | 136.3 ± 5.4 | 137.0 ± 5.7 | 137.2 ± 5.0 | 2.7 | 0.17 | **0.003** |
| Potassium, mmol/l | 4.1 ± 0.6 | 4.0 ± 0.5 | 4.1 ± 0.6 | 4.2 ± 0.6 | 4.3 ± 0.7 | 5.6 | **<0.001** | **<0.001** |
| Blood glucose, mmol/l | 7.7 ± 3.5 | 6.8 ± 2.6 | 7.4 ± 3.0 | 7.8 ± 3.4 | 9.3 ± 4.5 | 3.0 | **<0.001** | **<0.001** |
| BUN, mg/dl | 25.3 ± 18.0 | 19.3 ± 15.5 | 23.5 ± 15.9 | 27.7 ± 17.5 | 32.6 ± 20.8 | 2.9 | **<0.001** | **<0.001** |
| eGFR, ml/min/1.73m$^2$ | 82.6 ± 55.4 | 96.2 ± 38.4 | 85.0 ± 42.2 | 79.8 ± 83.1 | 65.4 ± 34.4 | 3.5 | **<0.001** | **0.03** |
| BNP, pg/ml | 274 (133–590) | 65 (37–183) | 155 (83–262) | 280 (154–592) | 480 (286–837) | 62.6 | **<0.001** | **<0.001** |
| Blood gas | | | | | | | | |
| pH | 7.41 (7.34–7.45) | 7.43 (7.39–7.46) | 7.42 (7.37–7.46) | 7.41 (7.34–7.45) | 7.36 (7.28–7.42) | 18.8 | **<0.001** | **<0.001** |
| PaO2, mmHg | 65.0 (56.0–79.0) | 64.0 (58.0–78.0) | 64.5 (56.0–79.5) | 64.0 (53.0–79.0) | 66.0 (57.0–79.0) | 19.0 | 0.68 | 0.57 |

*(Continued)*

**Table 1.** (Continued)

| | Global (N = 1,333) | Quartile I, <0.84 (N = 376) | Quartile II, 0.84–1.40 (N = 328) | Quartile III, 1.40–2.00 (N = 359) | Quartile IV, ≥2.00 (N = 270) | % missing value | p-value | Adjusted p-value* |
|---|---|---|---|---|---|---|---|---|
| | | | CSI quartiles | | | | | |
| PaCO2, mmHg | 40.4 (35.0–48.0) | 39.0 (34.0–44.0) | 41.0 (35.0–47.0) | 41.0 (35.2–51.0) | 42.0 (36.0–52.0) | 18.8 | <0.001 | <0.001 |
| Lactate, mmol/L | 1.10 (0.80–1.60) | 1.00 (0.70–1.50) | 1.00 (0.80–1.40) | 1.10 (0.80–1.70) | 1.10 (0.80–1.90) | 19.5 | 0.008 | <0.001 |

Values are mean ± SD, N (%), or median (25th and 75th percentiles).

*p-value adjusted for age and BMI at admission.

ACEi, angiotensin converting enzyme inhibitor; ARB, angiotensin receptor blocker; BMI, body mass index; BP, blood pressure; BPM, beats per minute; BUN, blood urea nitrogen; BNP, b-type natriuretic peptide; CSI, congestion score index; eGFR, estimated glomerular filtration rate; HF, heart failure; PaCO2, partial pressure of carbon dioxide; PaO2, partial pressure of oxygen; SD, standard deviation.

accuracy (AUC [95% CI] = 0.72 [0.75 to 0.81]), but the combination of CSI, BNP, and Brest score resulted in high diagnostic value for AHF (AUC [95% CI] = 0.85 [0.82 to 0.89]) (Fig 5).

## Validation in external cohorts

In the validation cohorts (N = 224), more than a half of the patients (56.7%) were men, mean age was 75.8 ± 13.9 years, mean CSI was 1.85 ± 0.87, and 72.7% had a diagnosis of AHF at discharge. The diagnostic performance of CSI was externally validated (AUC [95% CI] = 0.75 [0.68 to 0.82]).

## Discussion

Our results show that pulmonary congestion quantified by a simple standardized CXR scoring can efficiently identify patients with AHF in the ED, consistently across the various subgroups (e.g., elderly, overweight, COPD, and pneumonia). Furthermore, CSI significantly improved the reclassification of AHF diagnosis on top of the recognized clinical diagnostic markers of

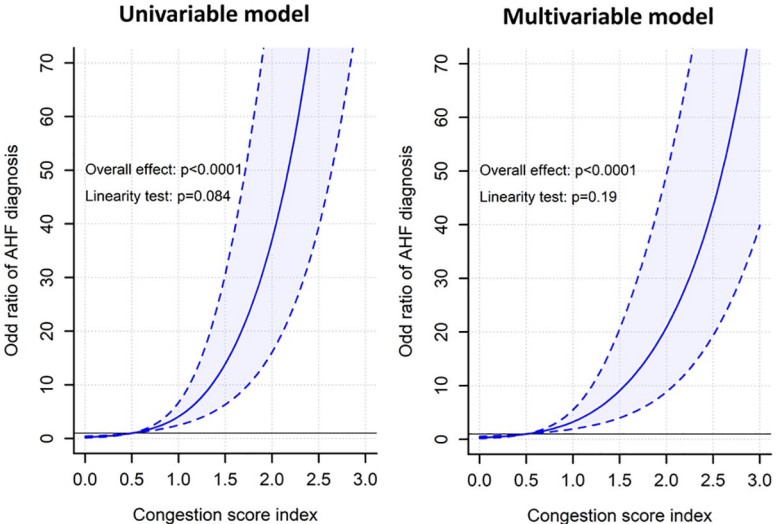

**Fig 2. Association between CSI and discharge diagnosis of AHF.** Multivariable model included the Brest score, which was calculated from age, comorbidity burden, dyspnea, ST segment abnormalities, and clinical congestion. Dotted lines/shaded regions represent 95% CI. AHF, acute heart failure; CSI, congestion score index.

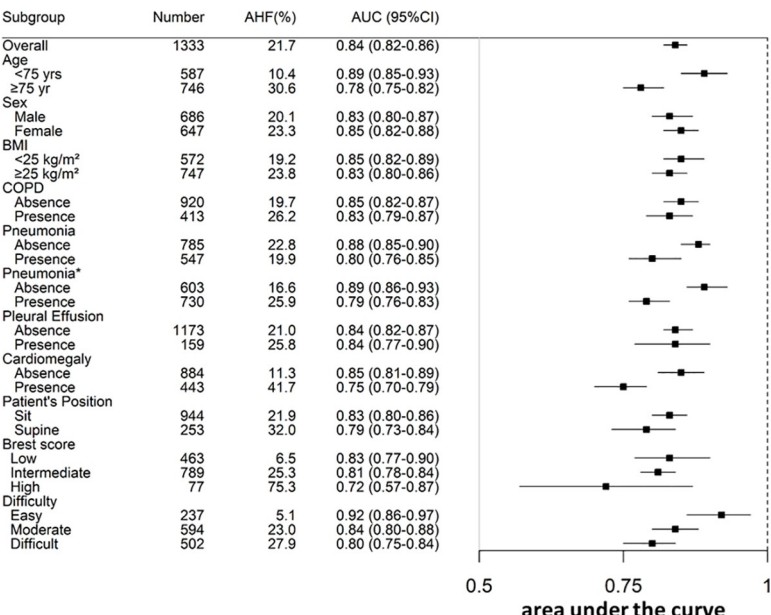

**Fig 3. Diagnostic value of radiographic CSI for AHF.** *Pneumonia was diagnosed at discharge. AHF, acute heart failure; AUC, area under the curve; COPD, chronic obstructive pulmonary disease; CSI, congestion score index.

AHF and natriuretic peptides. Our findings suggest that a semiquantified assessment of congestion on CXR could represent a readily available and clinically useful diagnostic tool for AHF in acute dyspneic patients in the ED.

## Radiographic congestion score index as a diagnostic tool for acute heart failure

CXR exhibited a higher diagnostic performance for AHF than usually reported in the literature [14–16]. Of note, previous reports assessed the diagnostic value of CXR using a single or combination of typical radiographic signs of congestion using a global evaluation of the lungs [15,16,31–33] rather than a systematic approach, with lung segmentation as used in the CSI method. Our group recently showed that more severe pulmonary congestion, quantified by either CSI or lung ultrasound at admission, was associated with higher pulmonary artery systolic pressure [19,34]. This association of CSI with hemodynamic data emphasizes the mechanistic plausibility of our results.

Recent registry data have shown that approximately 20% of patients hospitalized for AHF had concomitant lung diseases such as pneumonia and COPD [35–37], and these patients generally excluded in previous reports assessing the diagnostic value of CXR [16,17,38,39]. However, our subgroup analysis provided a remarkably homogenous diagnostic performance of CSI across various subgroups (i.e., elderly and BMI) and coexisting lung diseases (i.e., COPD and pneumonia). The only factor appearing to decrease the diagnostic value of CSI was cardiomegaly, possibly as a result of the overlapping of relevant information of the lung fields with cardiac silhouette in these patients.

## Radiographic congestion score index and other diagnostic measurements: Clinical parameters and natriuretic peptide

In keeping with previous literature data [19,40], our results showed that patients with more severe pulmonary congestion were predominantly elderly, had higher BMI, more

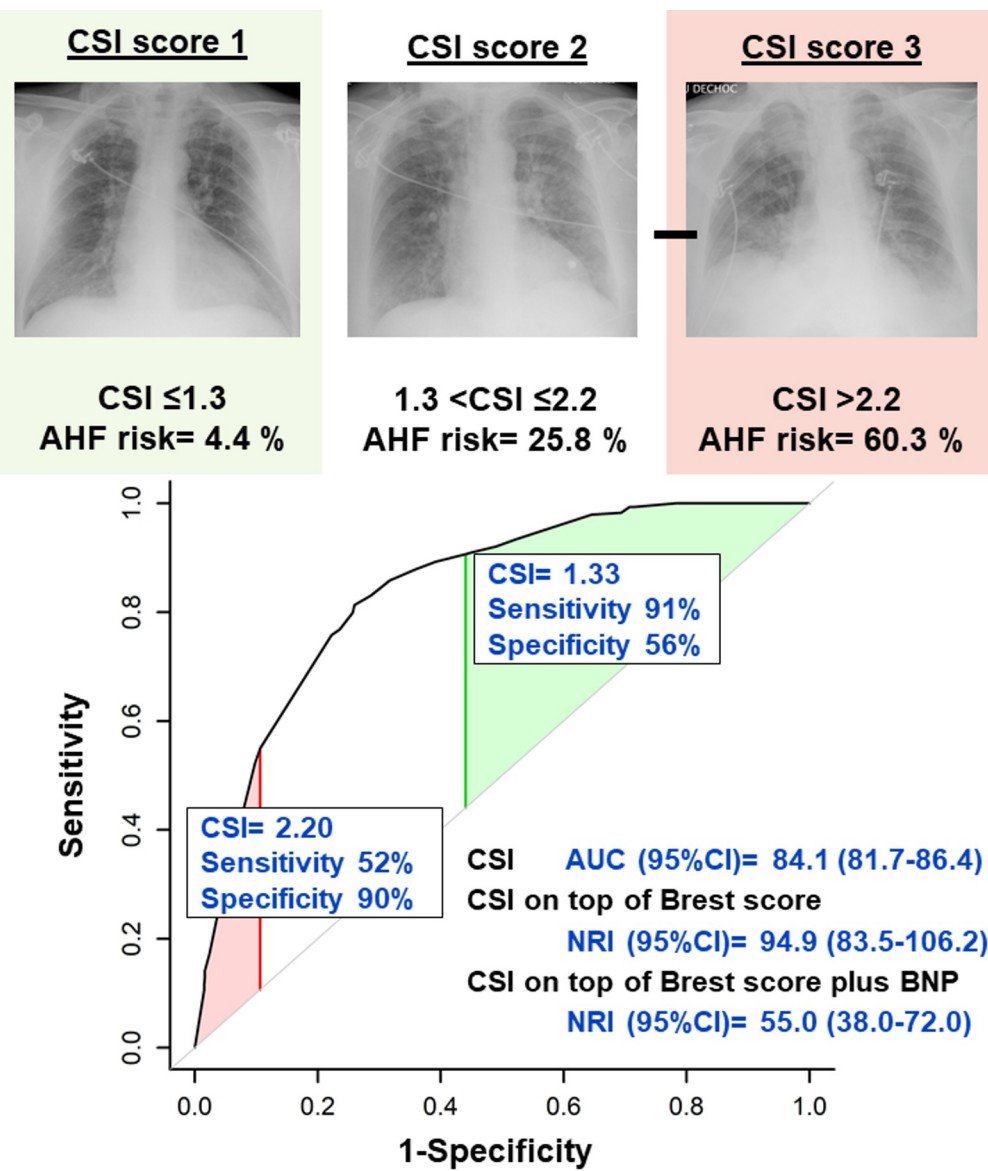

**Fig 4. Diagnostic performance of the radiographic CSI.** On the receiving operating characteristic curve for the association between CSI and AHF diagnosis, the red shaded region represents the 90% or greater specificity zone (CSI ≤ 1.3), whereas the green shaded region represents the 90% or greater sensitivity zone (CSI > 2.2). On the top portion of the figure, CXR panels illustrate typical examples of radiographies in the 3 zones, with CSI score of "1," "2," and "3," respectively, from left to right. AHF, acute heart failure; AUC, area under the curve; BNP, b-type natriuretic peptide; CSI, congestion score index; CXR, chest radiography; NRI, net reclassification improvement.

cardiovascular risk factors and comorbidities, severe congestion, poorer renal function, and higher inflammatory markers. In the latest guidelines, natriuretic peptide is recommended to rule out non-HF–related causes of acute dyspnea, although accumulated data suggested the overall diagnostic accuracy of natriuretic peptide [41–45]. However, multiple comorbidity burdens (i.e., older age and poor renal function) may lead to diagnostic uncertainty in a sizeable proportion of dyspneic patients [7]. BNP, in addition, requires time to measure and is not always available in routine clinical practice. In cases with unequivocal diagnosis of AHF based on clinical parameters, prompt treatment approach is recommended rather than wait for its

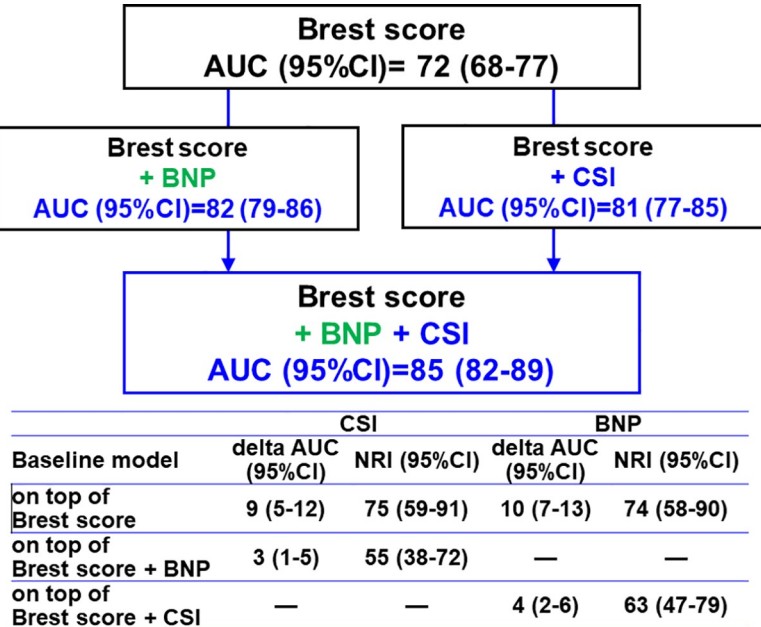

Fig 5. Added values of CSI and BNP for the diagnosis of AHF on top of the Brest score in patients with available BNP (*N* = 496). AHF, acute heart failure; AUC, area under the curve; BNP, b-type natriuretic peptide; CSI, congestion score index; NRI, net reclassification improvement.

result [6]. In this regard, the Brest score, based on clinical parameters, may be a pragmatic tool as well as good diagnostic accuracy for diagnosing HF as previously shown [27]. However, in the current study, more than half of dyspneic patients have intermediate scores, suggesting that this clinical score may need to be complemented by more refined strategies in this relatively frequent "grey zone." The unmet need in clinical practice, even when using the Brest score and natriuretic peptides, thus remains high.

In the current study, CSI was found to improve reclassification of AHF diagnosis on top of the Brest score and BNP. In addition, the combination of CSI and the Brest score improved the diagnostic value (AUC 0.81) to a similar degree of the combination of BNP and the Brest score (AUC 0.82). CSI also significantly improved diagnostic accuracy on top of the Brest score and BNP, and the combined used of these 3 parameters resulted in an AUC of 0.85. Taken together, these findings further strengthen that CSI may play a complementary role to the clinical model and natriuretic peptides in diagnosing AHF.

### Clinical implications

An early accurate diagnosis and consequently a prompt appropriate management improve outcomes in AHF patients admitted to the ED [5,46,47]. Our results show that a standardized evaluation of CXR, using CSI, improves diagnosis performance and potentially the ability to swiftly manage AHF in the ED. The assessment of radiographic pulmonary congestion requires training period. However, this approach may be easily scalable since training for this assessment is fairly simple; the operator (AD) who evaluated all CXRs of this cohort efficiently acquired the technique in the context of this study in a matter of a few hours.

Recent studies showed the clinical utility of lung ultrasound to diagnose AHF in dyspneic patients with high specificity and high sensitivity [9,26,48], and its diagnostic accuracy was better than that of CXR [8,50]. Of note, these previous studies did not include quantitative

assessment of radiographic pulmonary congestion. In any event, based on the promising results herein, multicenter trials may be warranted to assess the impact of the implementation of CSI on AHF diagnosis in patients with acute dyspnea. Furthermore, further study to compare diagnostic value between CSI on CXR and lung ultrasound is a worthy undertaking.

### Limitations and strengths

The main limitation of our derivation cohort was its single-center and retrospective nature, although the external validation of our results may lead to generalize our findings. The overall proportion of AHF diagnosis was relatively low, which may explain the fact that a low number of patients had congestion signs (i.e., leg edema, rales, and jugular venous distention) and cardiovascular diseases. In our center (as in most centers in France), some dyspneic patients known to have HF and all patients with obvious evidence of myocardial ischemia were admitted directly in the intensive cardiac care unit/cardiology ward, not through the ED [49]. The proportion of AHF diagnosis possibly due to the healthcare system may influence our results. However, it should be noted that hospitalization rates for worsening HF in the ED declined over the past decades [50], which may result from the development of a disease management program to prevent urgent HF hospitalization [51]. Indeed, the proportion of AHF diagnosis in the present study was similar to that of dyspneic patients in other contemporary cohorts [44,52]. Ejection fraction was not recorded; thus, we did not evaluate the diagnostic accuracy of CSI across levels of ejection fraction. This parameter, however, is usually not a major determinant of decision-making for acute dyspneic management in the ED [6,12].

In the derivation cohort, we had no data on CSI in 138 (9.4%) patients who did not undergo CXR or had no available lung field to assess CSI. These patients, however, had better clinical status (i.e., younger age and less severe congestion), and only 8% of these patients ($N = 11$) were diagnosed with AHF (S1 Table). In addition, all CSI readings were performed blinded for other parameters and diagnoses, suggesting that this limitation is unlikely to have major influence on our findings.

CSI is a semiquantitative tool with some subjectivity. Although accurate and reproducible scoring was achieved after about 3-hour training period in the current study and 1 of our previous study [19], more evidence may be needed to ascertain the appropriate learning period.

Lastly, the assessment of CSI was difficult in 502 (37.7%) patients, which may limit its applicability in routine clinical practice. However, the diagnostic value of CSI persisted in these patients (AUC = 0.80, 0.75 to 0.84) and the difficulty in assessing CSI did not influence its diagnostic accuracy ($P_{interaction} = 0.13$).

### Conclusions

Our study shows that a semiquantified assessment of radiographic pulmonary congestion provided diagnostic value for AHF in dyspneic patients of similar magnitude to that of BNP. These results suggest that implementing radiographic CSI in the diagnostic approach to AHF in addition to clinical parameters and BNP measurement could benefit the management of AHF patients. Better use of CXR may provide an inexpensive, widely, and readily available method for AHF triage in the ED. Multicenter prospective studies are nonetheless needed to confirm the diagnostic value of radiographic CSI.

### Supporting information

**S1 Checklist. TRIPOD checklist.**
(DOCX)

**S1 Fig. Flowchart.**
(DOCX)

**S1 Table. Baseline characteristics of patients with available and unavailable chest radiograph.**
(DOCX)

**S2 Table. Patient characteristics across different discharge diagnoses.**
(DOCX)

## Author Contributions

**Conceptualization:** Tahar Chouihed, Nicolas Girerd.

**Data curation:** Amine Douair, Déborah Jaeger, Gaetan Giacomin, Adrien Bassand, Victor Jeangeorges, Laure Abensur Vuillaume, Tahar Chouihed.

**Formal analysis:** Masatake Kobayashi, Kevin Duarte, Gregoire Preud'homme, Nicolas Girerd.

**Supervision:** Tahar Chouihed, Nicolas Girerd.

**Writing – original draft:** Masatake Kobayashi.

**Writing – review & editing:** Amine Douair, Déborah Jaeger, Gaetan Giacomin, Adrien Bassand, Victor Jeangeorges, Laure Abensur Vuillaume, Olivier Huttin, Faiez Zannad, Patrick Rossignol, Tahar Chouihed, Nicolas Girerd.

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
