## [Editor Report · Decision Letter 0]

1 Jun 2020

Dear Dr Kobayashi, 

Thank you for submitting your manuscript entitled "Diagnostic Performance of Congestion Score Index Evaluated from Chest Radiography for Acute Heart Failure in the Emergency Department" for consideration by PLOS Medicine.

Your manuscript has now been evaluated by the PLOS Medicine editorial staff [as well as by an academic editor with relevant expertise] and I am writing to let you know that we would like to send your submission out for external peer review.

Kind regards,

Caitlin Moyer, Ph.D.,

Associate Editor

PLOS Medicine

---

## [Decision Letter · Decision Letter 1]

15 Jul 2020

Dear Dr. Kobayashi,

Thank you very much for submitting your manuscript "Diagnostic Performance of Congestion Score Index Evaluated from Chest Radiography for Acute Heart Failure in the Emergency Department" (PMEDICINE-D-20-02033R1) for consideration at PLOS Medicine. 

[LINK]

In light of these reviews, I am afraid that we will not be able to accept the manuscript for publication in the journal in its current form, but we would like to consider a revised version that addresses the reviewers' and editors' comments. Obviously we cannot make any decision about publication until we have seen the revised manuscript and your response, and we plan to seek re-review by one or more of the reviewers. 

We expect to receive your revised manuscript by Aug 05 2020 11:59PM. Please email us (plosmedicine@plos.org) if you have any questions or concerns.

We look forward to receiving your revised manuscript. 

Sincerely,

Emma Veitch, PhD

PLOS Medicine

On behalf of Clare Stone, PhD, Acting Chief Editor,

PLOS Medicine

plosmedicine.org

*Please revise your title according to PLOS Medicine's style - ideally this should have the study design in a subtitle (after the main study question is summarised) after a colon - eg, "A randomized controlled trial," "A retrospective study," "A modelling study," etc.

*We'd ask the authors restructure the abstract using the PLOS Medicine headings (Background, Methods and Findings, Conclusions) - "methods and findings" should be a single subsection. 

*In the last sentence of the Abstract Methods and Findings section, please describe the main limitation(s) of the study's methodology.

*At this stage, we ask that you include a short, non-technical Author Summary of your research to make findings accessible to a wide audience that includes both scientists and non-scientists. The Author Summary should immediately follow the Abstract in your revised manuscript. This text is subject to editorial change and should be distinct from the scientific abstract. Please see our author guidelines for more information: https://journals.plos.org/plosmedicine/s/revising-your-manuscript#loc-author-summary

*Ideally, please reformat the in-text citations to use sequential numerals in square brackets (eg [1], [2] etc) rather than superscript numerals - if using referencing software this should be relatively straight forward.

*Did your study have a prospective protocol or analysis plan? Please state this (either way) early in the Methods section.

*The authors could consider using a reporting guideline designed to help enhance reporting of diagnostic studies, eg either STARD (https://www.equator-network.org/reporting-guidelines/stard/) or TRIPOD (https://www.equator-network.org/reporting-guidelines/tripod-statement/) depending on which is more appropriate to this study. If using one of the guidelines please complete and upload the appropriate checklist as supporting information alongside the revised paper (and consider citing the paper for the guideline in the methods section, noting that the tool was used to guide reporting of your study). 

Comments from the reviewers:

Reviewer #1: Authors reported and concluded that semi-quantified assessment of pulmonary congestion on chest radiography provided excellent diagnostic value for acute heart failure in dyspneic patients in the emergency settings. Furthermore, this assessment may be available method for acute heart failure triage in the emergency department. This finding is interesting, however, I have the following concerns.

1. Vascular congestion is the most obvious therapeutic target when approaching patients with acute decompensated heart failure. From the view point of evaluation methods to identify congestion, cardiac imaging including chest X-ray is highly needed. However, initial screening to diagnose congestion or acute heart failure is mainly performed by physical examinations in clinical practice; naturiuretic peptides is not routinely measured.

2. In this study, the signs and symptoms, including jugular venous distention, rales, and leg edema, are relatively low even in patients among Quartile IV. Although the authors indicated the accuracy of radiographic congestion index for acute heart failure, it may be unreasonable to conclude the utility of this index for acute heart failure triage in the emergency department.

3. Generally, the presence of pneumonia may obscure the appearance or degree of congestion in patients with decompensated heart failure. In this paper, the mean value of C-reactive protein was 6.6 mg/dL in global population, and no description of patients with clinically apparent pneumonia in the excluding criteria. And serum procalcitonin level could be useful to handle such patients, affecting key results.

4. They also concluded that the combination of congestion score index improved reclassification on top of the Brest score and BNP. However, the relationship between this index and prognosis should be presented.

5. In Methods, precise diagnosis of study population is unclear. Furthermore, selection bias may be present because the admission criteria in this study population is undecided.

6. In Methods, subjects receiving heart failure therapy and diagnosed "heart failure" at discharge during hospitalization due to other diseases were excluded? 

7. Reference 1 (published in 2006) may not be reflected current practice. 

8. The author selected various factors as model 1 in logistic regression analysis. These factors were previously shown as prognostic factors?

9. The authors described " an automated assessment of pulmonary congestion based on CSI could be developed・・・" in the discussion section. I may be a gap in their argument.

10. I wonder that a short time training of radiographic congestion score index is unconvincing.

11. The authors described "the large-sized population sample, the relatively homogenous management and adjudication of diagnosis" in the discussion section, there is no basis for these statements. 

Reviewer #2: I confine my remarks to statistical aspects of this paper. The basic approach is fine but I do have a few issues to resolve before I can recommend publication.

Abstract - What numbers come after the +- sign? (it seems like SD). Also, specify that the numbers in the parameter estimate are 95% CI

p. 6 Don't use quartiles. Use the raw score and, if desired, evaluate nonlinarity with a spline. Categorizing a continuous variable is almost always a mistake. Frank Harrell in *Regression Modelling Stragies* says "nothing could be more disastrous".

 Was collinearity evaluated?

p. 7 Last para - this doesn't look gradual! Also, see comment above.

Peter Flom

Reviewer #3: In this retrospective, single-center study, the authors evaluated the diagnostic value of Congestion Score Index (CSI) using a semi-quantitative method on chest X-rays. The authors conclude that the CSI is capable of improving heart failure prediction models in those patients who come to the emergency department with the diagnosis of acute dyspnea and that are based on the combination of clinical data (Brest scale) and natriuretic peptides (BNP).

The usefulness of chest radiography to evaluate patients with acute dyspnea in the emergency department is evident, however, the latest guidelines for heart failure recommend caution in their interpretation, since up to 20% of heart failure patients will not present radiographic alterations. (Eur J Heart Fail. 2016 Aug;18(8):891-975. doi: 10.1002/ejhf.592.). In addition, lung ultrasound has demonstrated its superiority over chest radiography for the evaluation of acute dyspnea, showing a better sensitivity and specificity values. (Crit Care Resusc . 2016 Jun;18(2):124., JAMA Netw Open. 2019 Mar; 2(3): e190703.).

Despite being a retrospective study, it has a good sample size and the statistical methodology is well developed. Furthermore, the authors validate their results in another cohort, which supposes an added value. However, despite being well executed methodologically, I think the following points should be clarified before being accepted: 

Comment 1: It is striking that only 10% of patients had a history of admission for heart failure. Furthermore, the percentage of cardiovascular diseases was relatively low despite the fact that the patients had an average age of 72 years. Do the authors believe that this fact could positively influence the results? In fact, in the group of patients with older age and in those who were more symptomatic (higher Brest scale), the AUC decreased the most.

Comment 2: If it is possible, it would be interesting to perform a sub-analysis of the ejection fraction (HFrEF vs. HFpEF). Patients with HFrEF usually debut with increased congestion and it would be interesting to see how CSI behaves in these subgroups.

Comment 3: As the authors well emphasize in the limitations section, up to 38% of the X-rays were difficult to interpret, I think it is an important fact, what was the degree of variability in the interpretation of the CSI in this group? If it is significant it should be taken into account. 

Comment 4: In the results and discussion section, it should be clear that the use of natriuretic peptides is useful because of their high negative predictive value. The degree of congestion (in this case measured by the CSI) should not be compared with the BNP concentration figures, BNP is not a good biomarker of congestion, this fact could explain the lower AUC.

Comment 5: Given that the authors acknowledge having experience in the use of lung ultrasound, it would be advisable to compare the results of this study, based on chest radiography, with other studies where lung ultrasound has been tested to identify patients with heart failure upon arrival at the emergency room. I think that such a comparison would make the discussion more attractive.

Comment 6: Figure 3 tries to include the AUCs of the different models, but in my point of view it is difficult to interpret. Probably a simpler table with the different variables included in each model and its AUC would be easier to interpret.

[LINK]

---

## [Decision Letter · Decision Letter 2]

21 Sep 2020

Dear Dr. Kobayashi,

Thank you very much for re-submitting your manuscript "Diagnostic Performance of Congestion Score Index Evaluated from Chest Radiography for Acute Heart Failure in the Emergency Department: An analysis from the PARADISE retrospective cohort" (PMEDICINE-D-20-02033R2) for review by PLOS Medicine.

I have discussed the paper with my colleagues and the academic editor and it was also seen again by two reviewers. I am pleased to say that provided the remaining editorial and production issues are dealt with we are planning to accept the paper for publication in the journal.

[LINK]

We look forward to receiving the revised manuscript by Sep 28 2020 11:59PM. 

Sincerely,

Caitlin Moyer, Ph.D.

Associate Editor 

PLOS Medicine

plosmedicine.org

Requests from Editors:

1. Response to reviewer 1: Point 3: Please do include the table “Patient Characteristics across Different Discharge Diagnoses” in the manuscript, if preferred this could be a supporting information file.

2.Data Availability Statement: “Under french law, deidentified data cannot be transferred to someone not authorized by the CNIL (comité national informatique et liberté) to perform the analysis. To comply with this national data reglementation, we will provide access to the data on a secured server held by our institution upon reasonable request to the primary investigator of the study. Importantly, Nancy CIC-P works with a number of international groups using this secured server.” At this time, please update your statement to include a web link or contact email address for data access. Please note that the contact cannot be one of the authors of the study. Please also capitalize the F in French.

3. Title: Thank you for revising your title. We suggest a minor change: “Diagnostic performance of congestion score index evaluated from chest radiography for acute heart failure in the emergency department: A retrospective analysis from the PARADISE cohort”

4.Abstract: Methods and Findings: Please provide the years during which the study took place and the setting/population of the individuals in the PARADISE cohort.

5. Abstract: Methods and Findings: Please include the p value for the following result: “CSI was significantly associated with acute HF diagnosis (adjusted-odds ratio for 0.1-unit CSI increase 1.19, 95%CI 1.16 to 1.22).”

6.Abstract: Methods and Findings: For the adjusted analyses presented, please include the important variables that are adjusted for in the analyses.

7.Abstract: Methods and Findings: For the following results, please remove subjective descriptions such as “excellent” and “good” and replace with quantitative terms. “The diagnostic accuracy of CSI for acute HF was excellent, whether alone [area under ROC curve (AUROC) 0.84, 95%CI 0.82 to 0.86] or in addition to clinical model (AUCROC 0.87, 95%CI 0.85 to 0.90). CSI improved diagnostic accuracy on top of clinical variables [Net reclassification improvement (NRI)=94.9%] and clinical variables plus BNP (NRI=55.0%). Good diagnostic accuracy was observed in the validation cohorts (AUROC 0.75, 95%CI 0.68 to 0.82).”

8.Abstract: Conclusions: Please replace the word “provides” with “provided” in the first sentence. Please remove the word “excellent” and replace with a more quantitative term. Please address the study implications without overreaching what can be concluded from the data; the phrase "In this study, we observed ..." may be useful.

9.Author Summary: The author summary should immediately follow the Abstract in your revised manuscript- although you have included it in the “response to reviewer/editor comments” section, it is missing from the manuscript.

10. Author Summary: What did the researchers do and find?: Please remove the word “strongly” as it is redundant: “This Congestion Score Index was significantly and strongly associated with acute heart failure diagnosis.”

11. Author Summary: What did the researchers do and find?: Please clarify to: “The Congestion Score Index also improved diagnostic accuracy over clinical parameters with or without inclusion of natriuretic peptide.” or similar.

12.Methods: Study Population: In the first sentence, please clarify what is meant by “consecutive patients”

13.Methods: Study Population: Please remove the trademark symbol from “Resurgences”

14.Methods: Study Population: Please specify the nature of participant consent, including whether informed consent was written or oral, or whether the requirement for participant consent was waived (and by whom).

15.Methods: Ethical approval: Please specify the nature of ethical approval obtained for your study (i.e. the outcomes reported here, including the validation cohort) rather than for the registered trial/ cohort.

16.Methods: Page 5: Thank you for your clarification that your study did not have a formal prospective analysis plan written. Please make sure that all pre-planned analyses are explicitly described as such, and note any changes in the analysis-- including those made in response to peer review comments-- in the Methods section of the paper, with rationale.

17.Methods: Page 6: Please note if the values in parentheses represent the confidence intervals, or other value: “Intra-class correlation coefficients showed good reproducibility [0.85 (0.71–0.93) and 0.81 (0.64–0.90) for intra and inter-observer reproducibility, respectively].”

18.Results: Page 8: For the following analysis, please present the p values to accompany the OR and 95% CIs: “...higher CSI showed an increased risk of AHF diagnosis [OR for CSI score 1.0=4.09 (2.50 to 6.71) , OR for CSI score 1.5=14.30 (6.50 to 31.82) , OR for CSI score 2.0 =37.03 (16.09 to 85.26) – considering CSI score 0.5 as reference]”

19.Results: Page 8: Please also present the ORs adjusted for Brest Score for the relationship between CSI and AHF diagnosis with CSI scores of 1.0, 1.5 and 2.0, as these aren’t easy to discern from the multivariable panel from Figure 2.

20.Results: Page 9: Please rename “Central illustration” as a numbered figure of the paper (for example, “Figure 5”)

21.Results: Page 10: Please revise this sentence to reflect the quantitative findings, rather than saying “the diagnostic performance of CSI was good” which is subjective.

22.Discussion: Page 13: Please change “provides” to “provided” and revise the Conclusion to avoid vague statements such as “excellent” diagnostic value- we suggest: “Our study suggests that a semi-quantified assessment of radiographic pulmonary congestion provided diagnostic value for AHF in dyspneic patients, similar magnitude to that of BNP. These results suggest that implementing radiographic CSI in the diagnostic approach to AHF in addition to clinical parameters and BNP measurement could benefit acute heart failure patients.” or similar. Please also add a sentence here that mentions that implementing a radiographic CSI in the diagnosis of AHF will require further evaluation, such as in a prospective study, before being applied in clinical practice.

23.Conflict of Interest: Please remove this section from the main text of the manuscript and instead provide this information in the relevant section of the manuscript submission form.

24.Acknowledgements: The information in this section is more appropriate for the “Funding” section of the manuscript submission form.

25.Figure 1: Please indicate in the legend what is indicated by a score of “0” and please augment the description in the legend to make it clear that these are examples illustrating the scoring system.

26.Figure 2: Please change “Odd ratio” to “Odds ratio” on the X axis (please indicate if there are missing decimal places). Please include in the legend all factors adjusted for in the mulitvariable model, and indicate (on the x axis) that these are the adjusted odds ratios. Please indicate that the dotted lines/shaded regions represent the 95% CIs.

27.Central Illustration: Please rename/number this figure. Please include a descriptive legend, describing what is shown in the radiograph panels, and defining the lines, shaded regions within the graph. Also, please indicate the nature of the values presented in parentheses alongside the AUC and NRI values in the figure.

28.Supplementary Table 1: Please include a legend and define abbreviations for AHF, ACEi/ARB, BP, BUN, eGFR, BNP, PH, PaCO2/PaO2

29.Checklist: Thank you for including the TRIPOD checklist in your response to reviewer/editor comments. However, the checklist does not seem to be included as a supporting information file. 

Please revise and submit the checklist, using section and paragraph numbers, rather than page numbers, to refer to locations in the text. 

Please add the following statement, or similar, to the Methods: "This study is reported as per the Transparent Reporting of a multivariable prediction model for Individual Prognosis Or Diagnosis (TRIPOD) guideline (S1 Checklist)."

Comments from Reviewers:

Reviewer #1: The manuscript has been improved by the revision.

I have no additional comments.

Thank you for giving a chance of re-reviewing the paper.

Reviewer #2: The authors have addressed my concerns and I now recommend publication

Peter Flom

[LINK]

---

## [Editor Report · Decision Letter 3]

14 Oct 2020

Dear Dr. Kobayashi, 

On behalf of my colleagues and the academic editor, Dr. Mitsutoshi Oguri, I am delighted to inform you that your manuscript entitled "Diagnostic performance of congestion score index evaluated from chest radiography for acute heart failure in the emergency department: A restrospective analysis from the PARADISE cohort" (PMEDICINE-D-20-02033R3) has been accepted for publication in PLOS Medicine. 

PRODUCTION PROCESS

Before publication you will see the copyedited word document (within 5 busines days) and a PDF proof shortly after that. The copyeditor will be in touch shortly before sending you the copyedited Word document. We will make some revisions at copyediting stage to conform to our general style, and for clarification. When you receive this version you should check and revise it very carefully, including figures, tables, references, and supporting information, because corrections at the next stage (proofs) will be strictly limited to (1) errors in author names or affiliations, (2) errors of scientific fact that would cause misunderstandings to readers, and (3) printer's (introduced) errors. Please return the copyedited file within 2 business days in order to ensure timely delivery of the PDF proof. 

If you are likely to be away when either this document or the proof is sent, please ensure we have contact information of a second person, as we will need you to respond quickly at each point. Given the disruptions resulting from the ongoing COVID-19 pandemic, there may be delays in the production process. We apologise in advance for any inconvenience caused and will do our best to minimize impact as far as possible.

PRESS

PROFILE INFORMATION

Thank you again for submitting the manuscript to PLOS Medicine. We look forward to publishing it. 

Best wishes, 

Caitlin Moyer, Ph.D.

Associate Editor 

PLOS Medicine

plosmedicine.org